electrical engineering/materials science/
power and energy systems

lithium-ion, characterization, performance
modelling, state of charge, thermal effect,
wireless sensor network

**Author for correspondence:**
Mohamad Khairi Ishak
e-mail: khairiishak@usm.my

# Battery characterization for wireless sensor network applications to investigate the effect of load on surface temperatures

Omer Ali[1,2], Mohamad Khairi Ishak[1], Chia Ai Ooi[1] and
Muhammad Kamran Liaquat Bhatti[2]

[1]School of Electrical and Electronic Engineering, Universiti Sains Malaysia, Nibong Tebal,
Pulau Pinang 14300, Malaysia
[2]Department of Electrical Engineering, NFC Institute of Engineering and Technology (NFC
IET), Multan 6000, Pakistan

OA, 0000-0002-0114-0457; MKI, 0000-0002-3554-0061

Wireless sensor networks (WSN) are commonly used in remote environments for monitoring and sensing. These devices are typically powered by batteries, the performance of which varies depending on environmental (such as temperature and humidity) as well as operational conditions (discharge rate and state-of-charge, SOC). As a result, assessing their technical viability for WSN applications requires performance evaluation based on the aforementioned stimuli. This paper proposes an efficient method for examining battery performance parameters such as capacity, open-circuit voltage (OCV) and SOC. Four battery types (lithium-ion, lithium-polymer, nickel-metal hydride and alkaline) were subjected to IEEE 802.15.4 protocol-based discharge rates to record the discharge characteristics. Furthermore, the combined effect of discharge rates on battery surface temperature and OCV variations was investigated. Shorter relaxation times (4–8 h) were observed in lithium-based batteries, resulting in faster energy recovery while maintaining rated capacity. It was observed that nearly 80% of the voltage region was flat, with minor voltage variations during the discharge cycle. Furthermore, lithium-based batteries experienced negligible changes in surface temperatures (approx. 0.03°C) with respect to discharge rates, making them the best battery choice for low-power applications such as WSNs.

# 1. Introduction

Wireless sensor networks (WSNs) have been used in a variety of application domains (residential, commercial and industrial) due to flexibility and ease of deployment. Due to the fact that these networks communicate wirelessly, they are simpler to deploy, control and maintain. Because of their adaptability, WSNs can be used both indoors and outdoors. Sensor nodes are usually deployed in inaccessible areas, and their battery life is usually a major concern. Several techniques have been proposed in the literature to increase sensor node and sensor network lifetime. The environmental conditions may have a significant impact on the behaviour of the sensor nodes in this perspective. The thermal effect, in particular, can have a major impact on the activity of the embedded platforms in sensor nodes [1,2] especially their batteries, which are extremely sensitive to temperature and load variations [3,4]. It is well established that the effective battery capacity of electrochemical cells varies with temperature [5,6]. The battery-powered systems are programmed to operate within a specified voltage range and to shut down when the cut-off voltages are reached. Several battery chemistries respond differently to environmental stimuli, particularly temperature changes and attached loads, and reach an early cut-off stage. As a result, predicting their behaviour over time is critical, particularly when designing energy-efficient algorithms that can extend device lifetime [7–10].

Energy optimization for WSN is a major topic due to limited energy availability and increased operational time requirements. Energy harvesting, energy transfer and energy conservation are the three main strategies used in WSN energy-efficient approaches. The first two approaches necessitate the employment of additional dedicated hardware to obtain energy for charging the on-board batteries. However, these strategies are not extensively used in large-scale WSN installations due to increased hardware complexity, cost and restricted energy transfer. As a result, at various architectural layers of WSN, energy conservation approaches are typically used to design energy-aware algorithms. Furthermore, most WSN networks are built and optimized using simulators to match real-world application behaviour in order to avoid high implementation costs and complexities. Practically all WSN simulators use simple mathematical models to implement the behaviour of batteries. This leads to inaccurate modelling that ignores environmental factors like load, temperature and humidity, culminating in significant inaccuracies. As batteries are highly nonlinear devices, with different discharge patterns depending on the battery chemistry, current SOC and operational conditions. As a result, designing energy-aware algorithms requires application-specific battery characterization for WSN.

Various battery chemistries have been introduced over the last three decades, each with its own unique specific energy, capacity, operating life and maintenance costs [11–13]. As a result, it is common practice to compare several cell chemistries for application-specific implementations in order to achieve the greatest techno-economic viability [14,15]. Furthermore, even under the same conditions, the batteries have contrasting energy profiles. This enables the system designer to select the appropriate battery type for their intended applications. For instance, alkaline batteries are heavily affected by load, thus estimating their capacity is extremely difficult. Load, temperature and state-of-charge (SOC) all have a significant impact on nickel-metal hydride (Ni-MH) and lead-acid batteries [16]. Lithium-ion batteries, on the other hand, have grown in popularity in recent years. These batteries have a higher specific energy, a low self-discharge rate, and lower operational and maintenance costs. Despite these advantages, lithium-ion batteries have some disadvantages [17]. They are fragile and require a safety circuit to monitor each cell's voltages and perform necessary regulation to fully use each cell's capability, further adding to system complexity and cost. The safety circuit limits some of the available energy during charge and discharge cycles by controlling cell voltages, thus reducing the available capacity.

The performance of battery chemistries in high specific energy and high-power situations (such as electric vehicles (EVs), renewable energy applications and micro-grid applications) has received a lot of attention [18–20]. These applications necessitate high currents from the batteries in a variety of conditions. However, WSN applications typically demand very low load currents that may change the battery discharge profile, and thus the device lifetime. Most of the WSN devices are designed using IEEE 802.15.4 protocol for low-power embedded devices [21].

In a typical wireless embedded system, the communication is achieved by a radio transceiver which is the most power-hungry component of all. IEEE 802.15.4 standard defines a low-power radio transceiver for efficient device connectivity to extend device longevity [22–24]. Because WSN nodes are deployed in large numbers, it is a common practice to employ WSN simulators to replicate their behaviour in the real world. This decreases operational complexity as well as the cost. However, the simulators use a linear energy source to model the batteries, which does not accurately reflect the real-world device

operation. These inaccuracies result in estimation errors of important intrinsic battery behaviour, which can affect overall sensor node lifetime [25–27]. Therefore, it is critical to examine the battery capacity characteristics under application-specific load profiles and environmental conditions in order to enable accurate energy-aware embedded systems.

Therefore, to overcome these limitations this research proposes a solution to characterize various battery chemistries for WSN applications. In this regard, four battery types were investigated in order to determine the best battery for WSN applications. Furthermore, a five-step comprehensive battery characterization strategy is proposed for accurate parameter extraction. To the best of our knowledge, no dataset on battery discharge characteristics for WSN applications is available. To address this issue, we used an efficient ampere-hour (Ah) approach to properly monitor battery discharge characteristics at different loads (IEEE 802.15.4 protocol-based transceiver drive profile). In addition, a comprehensive analysis on battery SOC and open-circuit voltage (OCV) is presented to report any hysteresis losses and capacity variations. Finally, to observe the thermal impact on the batteries, the effect of load currents on battery surface temperatures is also investigated, that may influence the device lifespan [28]. To this end, extended laboratory characterization has been performed to evaluate the battery parameters such as: capacity, OCV, SOC and their dependence on load profiles [29].

Some of the key findings of this research are presented as:

— the design of a systematic pre-conditioning test to remove battery passivation and to stabilize battery capacity;
— battery OCV calculations with varying relaxation time to estimate the thermodynamic stable states;
— investigating discharge characteristics of various battery chemistries following IEEE 802.15.4 radio transceiver load profiles;
— estimating battery hysteresis effect by comparing OCV and SoC;
— finally, investigating the effect of load currents on battery surface temperatures.

The rest of this paper is organized as follows. Section 2 discusses the related literature for battery characterization. Section 3 describes the experiment environment and five-step characterization methodology. Section 4 discusses the discharge characteristics of batteries along with the temperature effect on battery surface. Lastly, this research is concluded with key findings with final remarks.

## 2. Related Works

Coleman *et al.* [30], investigated the battery state-of-health (SOH) and SOC for lead-acid and lithium batteries. The discharge characteristics of these batteries were observed to determine battery health information. The proposed strategy used a two-discharge pulse approach, with the first pulse stabilizing the battery relative to its historical history and the second pulse estimating battery properties. This technique reduces discharge characterization time and can be used with a variety of battery chemistries. Along the same lines, Miniguano *et al.* [31], proposed a technique to accurately and efficiently extract battery performance parameters for EVs applications. The proposed method looked into several battery modelling and parameter estimation methodologies. The researchers investigated model accuracy versus simulation time, which may be applied to various battery chemistries.

Shateri *et al.* [32], proposed a machine learning-based technique to accurately estimate the SOC parameters for next-generation lithium-sulfur batteries. These batteries have a high energy specificity, a long lifespan and a modest ageing rate. The batteries have a fairly flat discharge curve, which makes estimating SOC for this discharge region difficult. The battery SOC was measured using a continuous real-time coulomb counting technique in their suggested scheme. The SOC data were then used to create a support vector machine (SVM)-based classifier model that properly estimated the SOC across a range of real-world load duty cycles. Stroe *et al.* [33], proposed a seven-step methodology to accurately characterize and model lithium-ion batteries. This technique provides a comprehensive, yet simplified laboratory characterization strategy for modelling battery discharge performance, which might be applied to several battery chemistries. The researchers properly predicted the battery performance parameters under different load profiles, which were then used to parametrize an analogous electrical model.

Ruffa *et al.* [34], investigated the thermal effect on battery packs for automotive applications. In this research, the effect of ambient and core battery temperatures were observed. The researchers argued that these critical factors can have a significant impact on battery charging and discharging and are crucial in

**Table 1.** Various battery chemistries used in characterization.

| tag | manufacturer | model | battery chemistry | capacity (mAh) | nominal voltage | C-rate |
|---|---|---|---|---|---|---|
| Batt1 | Powerizer [40] | MH-AAA1000APZ | nickel-metal hydride (Ni-MH) | 1000 | 1.2 V | 1 C |
| Batt2 | Data Power Technology [41] | DTP603450 | polymer lithium-ion (LiPo) | 1000 | 3.7 V | 1 C |
| Batt3 | Panasonic [42] | UF553443ZU | lithium-ion (Li-ion) | 1000 | 3.6 V | 1 C |
| Batt4 | Energizer [43] | LR-6 | alkaline (zinc, magnesium dioxide) | variable, load dependent | 1.5 V | 2 C |

extending the battery lifetime. Similarly, Christen *et al.* [35] explained the impact of temperature gradients on stability and lifetime of lithium-ion-based batteries. In their study, they developed a novel temperature gradient measurement technique that can properly predict the thermal properties of the battery and can also aid in the design of effective thermal management systems.

Over the last decade, there has been a huge surge in the integration of battery-powered wireless sensor networks and Internet of things (IoT) devices with digital systems. These devices have limited energy availability, and in most cases, the loss of battery energy marks the end of their useful life. As a result, much research has been conducted to explore the performance behaviour of batteries in different application domains. In this context, battery remaining energy, SOC and the effect of ambient temperatures are investigated in detail. Da Cunha *et al.* [36], studied the remaining alkaline battery energies for WSN application. The proposed technique used a low-cost hardware to monitor the battery remaining energy for both hardware as well as software resource utilization. It was observed that OCV may report inaccurate assessment of remaining battery capacity especially with inappropriate relaxation time. In addition, they also reported the limitation of current-sensing approach as it fails to measure the battery relaxation periods and may influence the discharge performance accuracy.

Similarly, Lajara *et al.* [37] proposed a node-based energy computational strategy for energy-aware WSN applications. A simple voltage and temperature measurement device monitors the parameters under various duty cycles, which is then used by a regression-based machine learning (ML) model to account for accurate SOC estimation. This scheme is suitable for low-cost, low-complexity deployment on micro-controllers. Furthermore, Rodrigues *et al.* [38] investigated the impact of environmental conditions on battery SOC and WSN node lifetime. A temperature-dependent battery model was incorporated in their proposed technique, parametrizing temperature and load profile impacts on a Ni-MH battery. The battery model calculates node lifespan reliably while accounting for temperature fluctuations and may be adapted to several battery chemistries.

# 3. Methodology

## 3.1. Experimental set-up

There is a broad range of battery chemistries on the market that are aimed at a variety of applications ranging from electric cars to grid integration and consumer electronics. These battery chemistries have various energy densities, specific energy, discharge cycles and costs [39]. Furthermore, while no particular battery is ideal, selecting the best battery chemistry for a certain application might increase the techno-economic viability. As a result, the study's battery characterization method is employed separately on all batteries while keeping device load profiles constant. The batteries used in this research are mentioned in table 1. Several battery chemistries with comparable capacity were used to maintain consistency. However, since Ni-MH and alkaline batteries have lower nominal voltages per battery, a pack of three cells was designed for each of these batteries to match the nominal voltage of other batteries. The batteries were mounted in a temperature-controlled chamber (Memmert HPC 410 Eco [44]) at room temperature during the characterization process, as shown in figure 1. A temperature sensor (PT100) mounted directly on top of the battery surface measured the battery surface temperature.

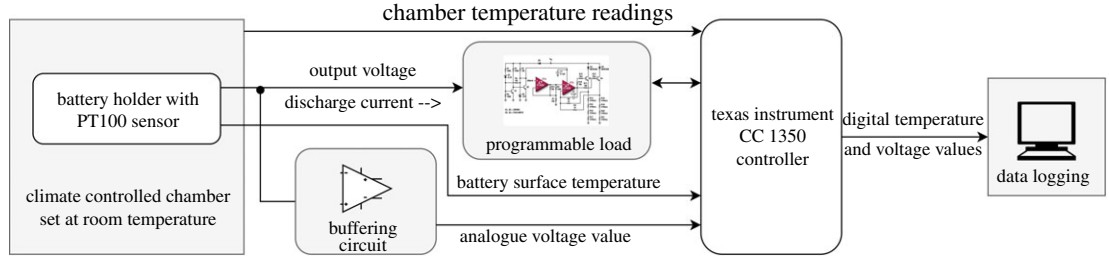

**Figure 1.** Test-bed schematic used for battery characterization and parameter extraction.

Environmental testing determines a component's capacity to perform in a certain environment. Extreme temperatures, humidity and other factors can all affect a device's failure. Batteries are the future of energy storage, and they must be tested to assure their safety and reliability. The batteries can fail due to undercharging, overcharging, overheating or a membrane break. Each of these failures carries a unique set of safety hazards that are specific to the product being tested. Batteries must be able to withstand extreme temperatures without leaking or exploding. Temperature fluctuations can have an adverse affect on cell voltage and can even reduce rated capacity. Thermal chamber testing can identify a battery's ability to tolerate severe temperatures and stress. Given the potential of battery failure, environmental chambers require safety features to prevent explosions and safeguard operators. Therefore, thermal chambers with thermal runaway detection are preferred to avoid explosion or fire hazards. In this perspective, the Memmert Eco Climate Test Chamber was used in this research that is designed to produce reproducible and accurate test results. Airflow systems that are optimized provide conditioning throughout the workspace, decreasing gradients and improving consistency. Application-specific heating enables tailored temperature change rate performance to match the needs of individual testing.

These tests used discharge currents of 20, 30 and 50 mA to simulate the working states of the IEEE 802.15.4 protocol-based CC2420 radio transceiver, which is commonly used in WSN applications. These values correspond to discharge rates of 0.02, 0.03 and 0.05 C, respectively. The C-rate is commonly used to investigate charge and discharge currents in relation to battery capacity. On average, based on the chosen C-rates, the resulting experiments had the duration of approximately (28, 23, 17 and 3.5 h) per battery. The Texas Instruments CC1352R2 development board was used to configure the discharge current profiles and record the battery statistics.

The remote sensing, data aggregation and information transmission are common requirements for WSN applications. To save energy, practically every situation avoids continuous or real-time communication. As a result, WSN nodes are typically operated in duty-cycling mode to preserve energy and extend device lifetime. As a result, different data transmission rates are generated, which are designed based on the intended use. These data transfer rates can range from low to medium to high to near real-time. However, in these tests, we maintained the data rate modest in order to provide recuperation times between discharge pulses. As a result, every 15 s, a discharge pulse enabled the load, and the analogue voltage changes were then converted to digital using the on-board 12-bit analogue to digital converter (ADC). This circuit can record current variations ranging from 50 A to 1000 mA, making it suitable for recognizing discharge patterns with minimal offset and noise. The current consumption profiles for a typical CC2420 radio transceiver are presented in table 2.

## 3.2. Battery characterization

The primary purpose of the suggested characterization methodology was to acquire performance parameters for the aforementioned battery chemistries for application-specific load profiles. Their reliance on load currents, as well as the effect on battery surface temperature, were also examined. Figure 2 illustrates the proposed methodology steps that were used in this analysis. The pre-conditioning and relaxation tests were required prior to each discharge process, hence they are addressed here, whereas the remaining tests are discussed in the part that follows.

### 3.2.1. Pre-conditioning test

For battery testing, current is usually expressed proportional to the battery capacity. The capacity, on the other hand, is not a fixed value and is strongly influenced by the discharge rates. A constant current

**Figure 2.** Test sequence for battery characterization.

**Table 2.** Current consumption profiles used in battery characterization.

| working state | value | unit |
|---|---|---|
| CC2420 radio transceiver current consumption | | |
| *transmission state (Tx)* | | |
| power = −25 dBm | 8.5 | mA |
| power = −15 dBm | 9.9 | mA |
| power = −10 dBm | 11 | mA |
| power = −5 dBm | 14 | mA |
| power = 0 dBm | 17.4 | mA |
| *receive state (Rx)* | | |
| power = 0 dBm | 18.8 | mA |
| active mode (at 48 MHz) | 2.9 | mA |
| active mode (at 48 MHz) while using onboard sensors | 14.2 | mA |

discharge is commonly used to discharge the battery in a certain number of hours. This is the concept of rated potential $Cn$, where $n$ represents the time base. For accurate comparison of test outcomes, the cell potential of all measured cells must be calculated in the same way. In this study, the pre-conditioning approach serves two purposes. First, it enables the removal of any passivation or memory effect in cells. Second, a sequence of charge and discharge cycles stabilizes the battery capacity. This technique aids in maintaining electrolyte porosity during the first few high-current discharge cycles. Five consecutive charge and discharge cycles at $1 - C$ values (1 A) at room temperature were used in the pre-conditioning tests [45]. Alkaline cells, on the other hand, were not pre-conditioned because they were not rechargeable and there was no knowledge of their nominal capacity. Figure 3 discusses the pre-conditioning of the batteries used in this experiment (further referred to using their Tag names as given by table 1).

The lithium-based cells were found to have a reduced capacity at first, but after the third iteration, the battery capacity was stabilized. The measured output demonstrates a small monotonic rise in battery capacity after stabilization. Furthermore, it can be demonstrated that the lithium-based cells exhibit less than 4% heterogeneity across two consecutive discharge cycles. Furthermore, lithium-based batteries showed a stabilized capacity of 960 mAh, which is 4.16% less than the rated capacity. The Ni-MH cells, on the other hand, showed a strong monotonic rise in capacity throughout the experiment. Despite this, capacity changes between discharge cycles were less than 3%. In comparison with lithium-based chemistries, Ni-MH chemistries exhibited lower battery capacity due to higher self-discharge rates.

### 3.2.2. Relaxation test

Internal battery phenomenon such as polarization and mass transfer occur during the rest period and have very slow dynamics resulting in voltage fluctuations. The relaxation phase is the period

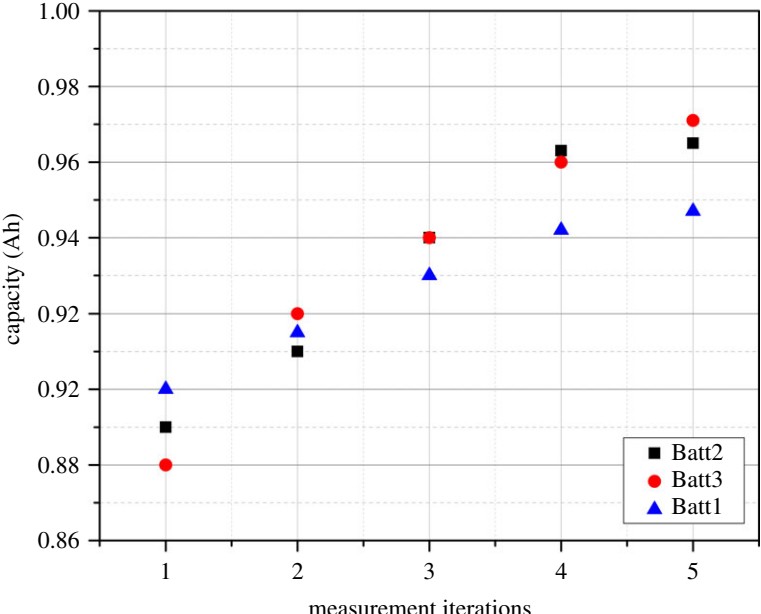

**Figure 3.** Discharging capacity of batteries during preconditioning tests.

following discharge during which there is no load attached and the battery voltages gradually approach a stable state [46,47]. Precise OCVs at various battery SOCs are needed to accurately estimate the battery SOC at any given time. As a result, in a no-load condition, a relaxation time must be introduced between voltage measurements to enable the battery to achieve thermodynamic stability. The longer the relaxation period, the closer to the actual OCV is the battery voltage. In reality, after several hours the battery voltage is still changing. In this regard, techniques such as electrochemical impedance spectroscopy (EIS) are deployed to measure minute changes in battery impedances during relaxation phase. In [48], the battery voltage could be stabilized beyond 24 h relaxation period. Many studies found that after 5 h, the change in battery impedance had slowed dramatically. However, it is uncertain whether it has slowed considerably enough that any future impedance changes are inconsequential [49–52].

However, no single relaxation period that could be applied for different battery chemistries has been documented. To that end, the batteries were fully charged before being discharged at three separate SOC levels (corresponding to 90%, 50% and 10%, respectively). For each set of battery discharge trials, a variety of specified relaxation times (10 min, 1, 2, 4, 8 and 24 h) were used. During this relaxation time, voltages were measured with a 1 min resolution and were compared with OCV measurements. The batteries were considered quasi-stabilized at this stage with negligible voltage variations within relaxation stage. These trials with varying relaxation times assisted us in determining the ideal relaxation period that may be used in future low-current discharge experiments. The OCV error between stabilized battery states can be computed from equation (3.1).

$$\delta_{\text{OCV}} = \frac{|V_{\delta}t - \text{OCV}|}{\text{OCV}} \times 100, \tag{3.1}$$

where $\delta_{\text{OCV}}$ and $V_{\delta}t$ represent OCV error and measured voltages over time intervals. The obtained results, along with the OCV errors are reported in table 3.

It was observed that lithium-based battery chemistries are greatly influenced by shorter relaxation periods, where the OCV voltages are significant. The computed errors are slightly larger between 10 min and 4 h intervals. However, the batteries appear to be quasi-stabilized between 8 and 24 h, exhibiting a minimum change. On the other hand, Ni-MH-based battery showed a slightly lesser variation in OCV measurements, reaching steady state within 4 h. Thus, a relaxation time of 8 and 4 h were chosen for lithium and Ni-MH-based batteries, respectively. These relaxation periods were later used to characterize the battery discharge performances.

**Table 3.** Recorded voltage measurements at SOC (10%, 50% and 90%) after relaxation period.

| relaxation time | SOC (10%) | | SOC (50%) | | SOC (90%) | |
|---|---|---|---|---|---|---|
| | $V_\delta t$ (V) | $\delta_{OCV}$ (%) | $V_\delta t$ (V) | $\delta_{OCV}$ (%) | $V_\delta t$ (V) | $\delta_{OCV}$ (%) |
| **Li-ion (Batt3)** | | | | | | |
| 10 min | 3.68 | 5.154 | 3.7 | 4.639 | 3.73 | 3.865 |
| 1 h | 3.72 | 4.123 | 3.75 | 3.350 | 3.78 | 2.577 |
| 2 h | 3.758 | 3.144 | 3.792 | 2.268 | 3.81 | 1.804 |
| 4 h | 3.798 | 2.113 | 3.816 | 1.649 | 3.847 | 0.850 |
| 8 h | 3.867 | 0.335 | 3.871 | 0.231 | 3.875 | 0.128 |
| 24 h | 3.88 | 0 | 3.88 | 0 | 3.88 | 0 |
| **Li-Po (Batt2)** | | | | | | |
| 10 min | 3.71 | 5.597 | 3.76 | 4.325 | 3.79 | 3.562 |
| 1 h | 3.771 | 4.045 | 3.79 | 3.562 | 3.801 | 3.282 |
| 2 h | 3.891 | 0.992 | 3.847 | 2.111 | 3.847 | 2.112 |
| 4 h | 3.899 | 0.7888 | 3.882 | 1.221 | 3.889 | 1.043 |
| 8 h | 3.92 | 0.254 | 3.921 | 0.229 | 3.93 | 0 |
| 24 h | 3.93 | 0 | 3.93 | 0 | 3.93 | 0 |
| **Ni-MH (Batt1)** | | | | | | |
| 10 min | 3.842 | 3.467 | 3.871 | 2.738 | 3.879 | 2.537 |
| 1 h | 3.876 | 2.613 | 3.882 | 2.462 | 3.894 | 2.160 |
| 2 h | 3.934 | 1.155 | 3.883 | 2.437 | 3.929 | 1.281 |
| 4 h | 3.9718 | 0.226 | 3.973 | 0.175 | 3.979 | 0.025 |
| 8 h | 3.98 | 0 | 3.98 | 0 | 3.98 | 0 |
| 24 h | 3.98 | 0 | 3.98 | 0 | 3.98 | 0 |

# 4. Results and discussions

## 4.1. Battery capacity test

The capacity of a battery is affected by the applied load as well as the operating temperature. Furthermore, battery capacity degrades over time, especially with deep charge and discharge cycles. However, since the end of life of WSN nodes is determined by a fully discharged battery, the battery SOH and ageing are beyond the scope of this work. The capacity test was designed to approximate discharge capacity under different load profiles (0.02, 0.03 and 0.05 C). To avoid the influence of ambient temperatures on the batteries, these experiments were carried out at room temperature in a climate chamber. The dependance of discharge capacity on C-rates is reported in figure 4. It was observed that the discharge capacity of all the batteries decrease with the addition of load. Other than alkaline cells, the discharge profile of the batteries shows a small monotonic decrease in capacity. This effect was investigated further by measuring the Peukert coefficient, which is given by equations (4.1) and 4.2.

$$t = H\left(\frac{C}{IH}\right)^k \tag{4.1}$$

and

$$C_p = I^k t, \tag{4.2}$$

where $t$, $H$, $C$, $I$, $k$ and $C_p$ represent time in hours, the discharge time based on ampere-hour specifications, battery capacity, discharge current, Peukert exponent and nominal discharge capacity at 1 Ah, respectively.

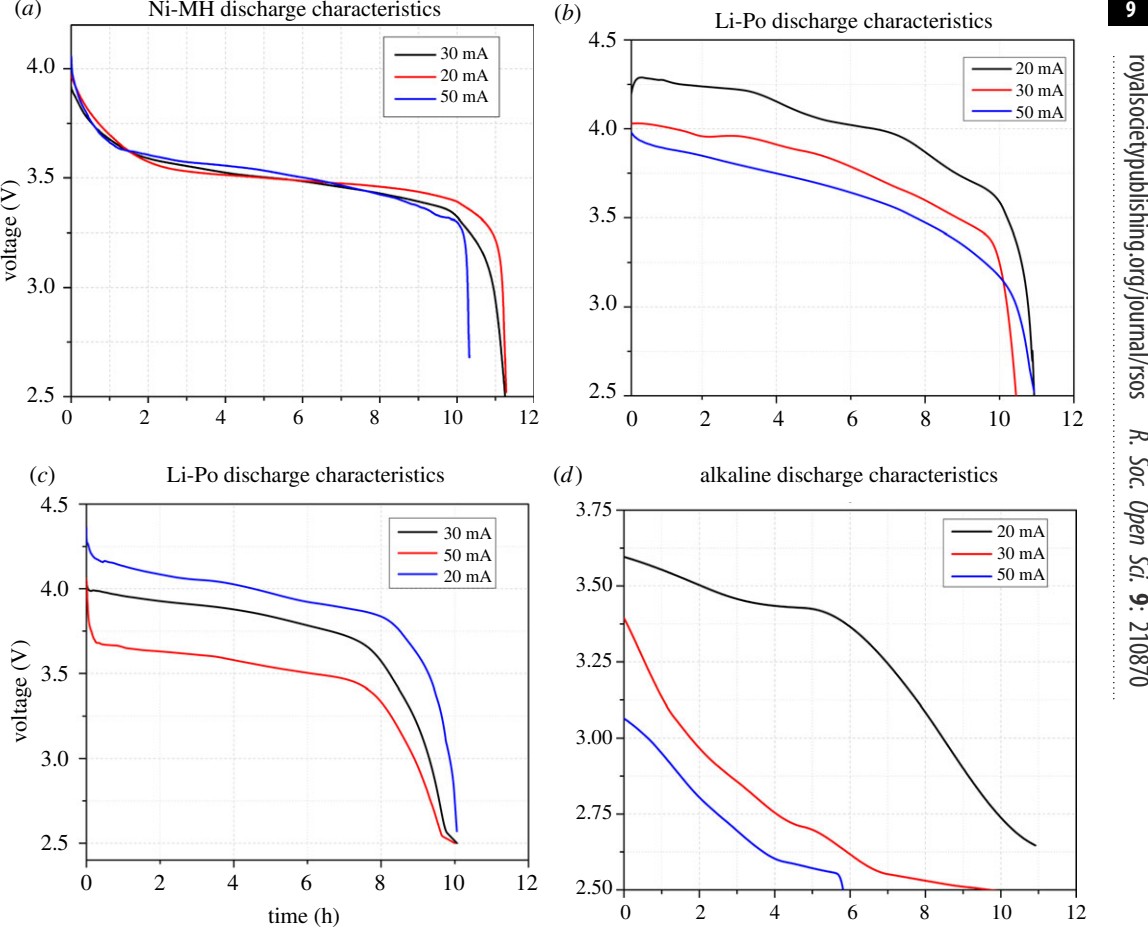

**Figure 4.** Discharge characteristics of the batteries under load current profiles. (a) Nickel metal hydride Ni-MH (Batt1), (b) lithium polymer Li-Po (Batt2), (c) lithium ion Li-ion (Batt3) and (d) zinc magnesium dioxide, alkaline (Batt4).

The Peukert number was found to be close to 1 for lithium-ion and lithium-based batteries, exhibiting almost less than 1% reduction in discharge capacity. However, Ni-MH battery reported a Peukert number between 1.4%, which is in agreement with capacity reduction as the load increases. Furthermore, it is important to consider that the Peukert Law does not account for self-discharge rate, which was found negligible after each relaxation period.

Ni-MH-based battery exhibited almost 80% flat region for 0.02 and 0.03 C discharge rates. However, its capacity reduces as the load is increased. The nickel-metal hydride battery's discharge behaviour is generally well suited to the needs of today's consumer devices, particularly those demanding a constant voltage for prolonged periods of operation or rapid rate discharge. The capacity attained from a new, but properly conditioned, battery submitted to a constant-current discharge at room temperature from being optimally charged is measured in terms of C-rate, that helps to normalize capacity calculations. Because battery capacity varies inversely with discharge rate, capacity ratings are affected by the discharge rate. Due to the general wound construction, improved connections and wide surface area of the electrodes, Ni-MH batteries have a comparatively low internal resistance (IR). Because of the low battery IR, Ni-MH batteries have outstanding high-rate performance.

The environmental parameters, particularly discharging temperature and discharge rate, have an impact on the discharge voltage profile. Under most conditions, however, the voltage curve maintains a flat steady state desired for electronics applications. This explains the flat zone in the discharge curves. The nickel-metal hydride battery, like lithium batteries, has a steep 'knee' near the conclusion of the discharge where the voltage drops quickly. Furthermore, for rates less than 1 C, there is no noticeable effect on the shape of the discharge curves; for rates greater than 1 C, both the beginning and ending transients consume a greater amount of the discharge length. Finally, it was discovered that discharge rates less than 0.5 C have no meaningful effect on the capacity. On the other hand,

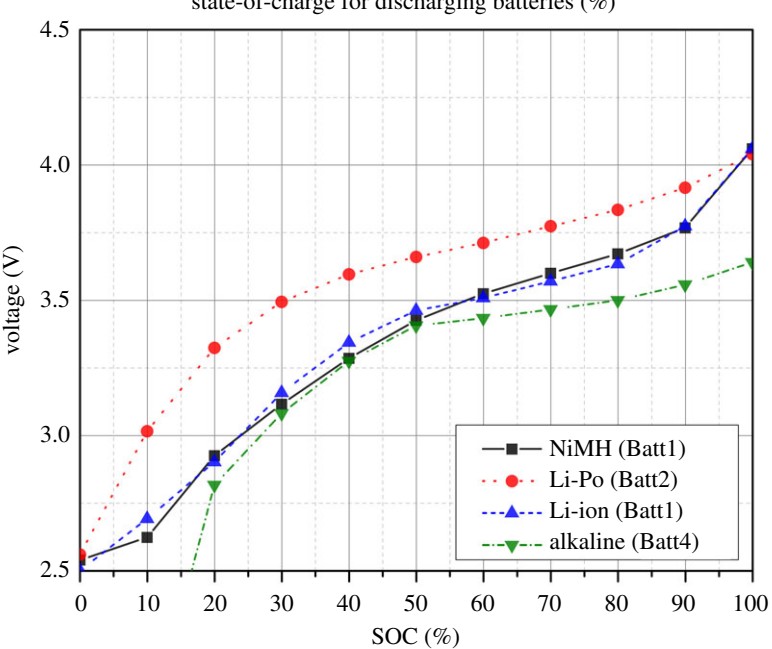

**Figure 5.** OCV measurements against battery discharging SOC.

significant voltage reductions occurred at discharge rates of 0.5 C and above. Depending on the discharge termination voltage selected, this voltage drop may also result in capacity reduction.

Moreover, both lithium-based battery chemistries provided a significantly flat region while maintaining the capacity. Lithium-polymer-based battery suffers slightly in capacity reduction as the load is increased when compared with lithium-ion-based chemistries. This load dependency has a more pronounced effect at higher loads, whereas for WSN application load profiles, the capacity reduction is marginal. Alkaline cells on the other hand, have a sloping discharge profile that is strongly dependent on the applied load. It was also observed that the battery capacity for alkaline-based cells decreases significantly as the applied load is increased. In addition, these batteries provide a highly nonlinear region for most part of the discharge curve, making available energy, SOC and capacity estimations almost impossible.

A small cell pack is typically designed for Ni-MH and alkaline-based batteries in low-power applications. On one hand, this allows for the provision of the required device voltages; on the other hand, it necessitates charge balancing, which is both difficult and inefficient for WSN applications. As a result, when a single cell exceeds cut-off voltage, the entire battery pack shuts down, resulting in a premature battery cut-off. Accurate estimation of available battery energy and its discharge profile is very important for system designers. As WSN are powered by limited energy, SOC estimation plays a crucial role in optimizing the WSN node for energy-aware applications. Since the supply voltage remains relatively constant during the discharge period, a flat discharge curve simplifies the configuration of the device in which the battery is used. The power delivered by cells with a sloping discharge curve decreases gradually over the discharge period. This could cause issues for high-power applications at the end of the cycle. If the slope is too steep for low-power applications that need a steady supply voltage, a voltage regulator can be needed, which would incur additional cost and power consumption.

## 4.2. Open circuit voltages versus state-of-charge

OCV, like most electrochemical battery parameters, is affected by operating conditions such as temperature, SOC, etc. As a result, the primary aim of these experiments was to determine the OCV dependence on battery SOC while maintaining a constant temperature during discharge. The OCV was calculated in 5% increments using the methods described in [53] and is reported in figure 5. The OCV is affected by thermodynamical processes in addition to its usable ability. Overvoltage and OCV hysteresis found during charging and discharging are attributed to battery losses. OCV

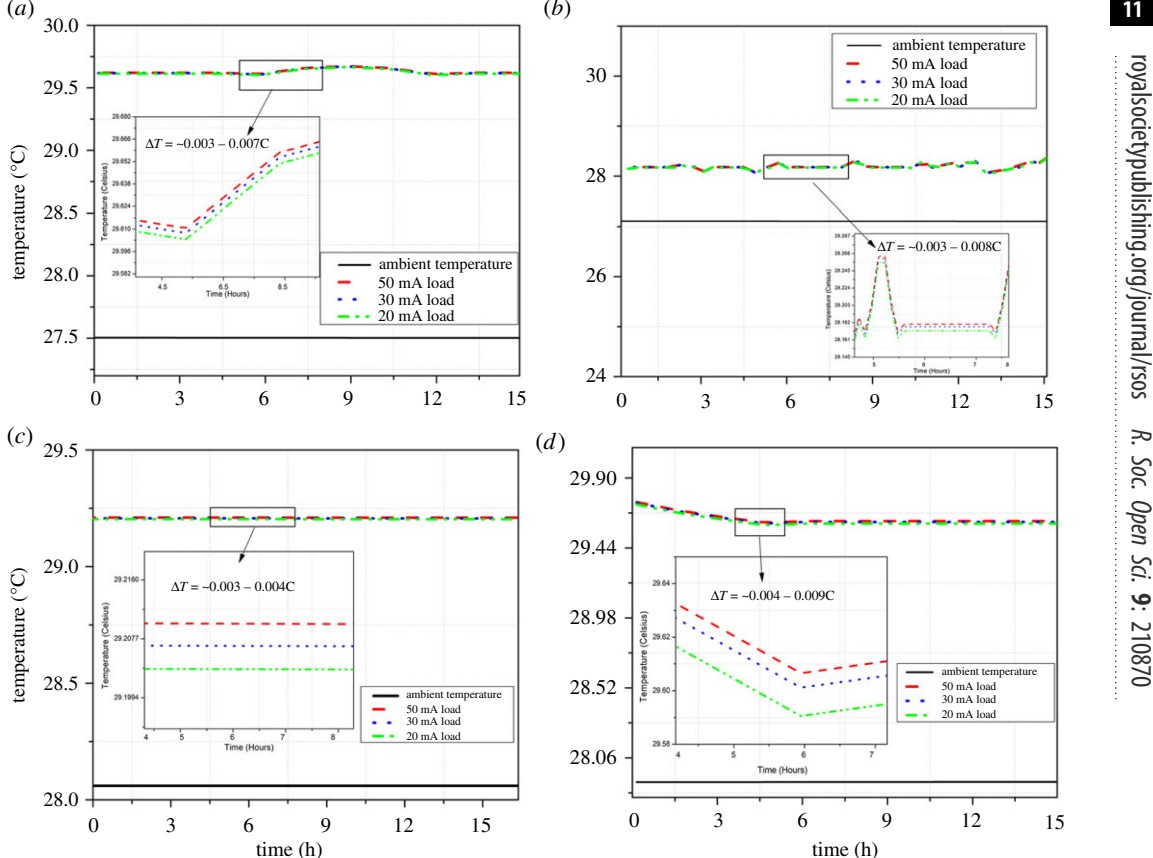

**Figure 6.** Effect of Load currents on battery surface temperatures. (*a*) Nickel metal hydride Ni-MH (Batt1), (*b*) lithium polymer Li-Po (Batt2), (*c*) lithium ion Li-ion (Batt3) and (*d*) zinc magnesium dioxide, alkaline (Batt4)

hysteresis is a common occurrence in Ni-MH and lithium-ion batteries. Thermodynamic entropic effects, mechanical stress and microscopic distortions within the active material particles all contribute to hysteresis [54].

Lithium and Ni-MH batteries exhibited hysteresis around their flat SOC area. This is mostly due to the inherent structure of active battery materials, which undergo chemical transformations during operation. The hysteresis effect is primarily investigated in high-precision, high-power battery control systems that involve charge balancing. However, it is largely neglected in WSN implementations for two reasons. First, the hysteresis impact is minimal. Second, since the batteries are mostly used once, complicated and costly charge balance circuits are avoided.

Furthermore, OCV measurements are critical for estimating voltage-based SOC. Each battery chemistry has its own discharge fingerprint. Although voltage-based SOC works effectively for a resting lead-acid battery, the flat discharge curve of nickel- and lithium-based batteries renders it ineffective. The discharge voltage curves of Li-ion, Li-Po and Ni-MH batteries are relatively linear, with almost 80% of the stored energy remaining in the flat voltage profile. While this trait is appealing as an energy source, it complicates battery SOC charge assessment in the flat zone since a minor change in voltage may imply a bigger change in estimated SOC.

## 4.3. Battery surface temperature under load

A substantial amount of study has been undertaken to quantify the thermal effect of applied load on batteries. High-power and high-energy applications, such as EVs, solar energy and grid storage, show an increase in battery temperature.

The majority of these thermal effects are studied in the laboratory by evaluating the core internal battery temperatures. By contrast, the majority of low-cost consumer products use battery surface temperature to regulate thermal impact and trigger early warnings before thermal runaway. The effect

of temperature influence on batteries as a result of a lower attached load, on the other hand, has not been reported. As a result, the goal of this test was to investigate the variations in battery surface temperature for different load profiles.

In this regard, all of the batteries were evaluated at ambient temperatures inside environment-controlled chambers with surface-mounted temperature sensors. The batteries were individually depleted at 0.02, 0.03 and 0.05 C to examine temperature changes on the surface of the batteries. The recorded temperature variations on battery surfaces are presented in figure 6. The batteries were fully discharged, and the surface temperature differences for each load profile were reported. It was observed that all of the batteries had a small temperature rise when compared with the ambient temperature. Throughout all of the trials, lithium-ion batteries showed a slight temperature increase of (0.003−0.004°C).

Li-Po- and Ni-MH-based batteries, on the other hand, showed minor monotonic temperature differences of 0.003°C and 0.008°C, respectively. Alkaline batteries behaved similarly, with temperature differences ranging from 0.004°C to 0.009°C. This increase in battery surface temperature is insignificant and represents no increase in internal temperature caused by the attached load. As a result, it was determined that lower currents, equivalent to WSN application load profiles, can have little effect on internal battery temperatures. This simplifies the calculation of battery energy for low-power applications where it is solely dependent on load and SOC.

The tests have shown that the influence of these attached loads can be totally overlooked for high-precision low-power SOC applications. The ambient temperature, however, can vary greatly due to the effect of WSN deployments (−20°C to 60°C). As a result, for reliable SOC measurement only the influence of ambient temperature on battery power and OCV must be investigated.

# 5. Conclusion

In this paper, we presented a detailed battery characterization and parameter extraction method for small load applications. In this regard, four battery chemistries were investigated in order to determine the best battery for WSN applications. The pre-conditioning tests were carried out to remove battery passivation and stabilize battery capacity, recovering nearly 4% of the battery capacity. Furthermore, the relaxation tests enabled the calculation of the optimum time for battery energy recovery for all of the battery types used in this study. Lithium-ion batteries provided higher voltages and larger flat regions across all discharge rates. Furthermore, the thermal effect of higher discharge rates on lithium-based batteries was found to be negligible, making these batteries the best choice for WSN applications.

Data accessibility. Battery characterization dataset available at Dryad doi:10.5061/dryad.g1jwstqr4.

Authors' contributions. O.A. conceived and designed the experiments, performed the experiments, analysed the data, performed the computation work, prepared figures and/or tables, authored and compiled the final draft. M.K.I. conceived and designed the experiments, analysed the data, reviewed drafts of the paper and approved the final draft. C.A.O. designed the experiments, analysed the data, performed the computation work, reviewed drafts of the paper and approved the final draft. M.K.L.B. analysed the data, reviewed drafts of the paper, and approved the final draft.

Competing interests. We declare we have no competing interests.

Funding. This work was sponsored by Universiti Sains Malaysia (USM), Research grant no. RUI:1001/PELECT/8014049.

Acknowledgements. The authors thank Dr Maham Hussain (https://www.scopus.com/authid/detail.uri?authorId=57193433139) (Department of Chemical Engineering, NFC Institute of Engineering & Technology Multan) for her support during the review of this article.

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
