## [Peer Review File · Royal Society Open Science]

Review History

RSOS-210870.R0 (Original submission)

Review form: Reviewer 1

Is the manuscript scientifically sound in its present form?

Yes

Are the interpretations and conclusions justified by the results?

Yes

Is the language acceptable?

Yes

Do you have any ethical concerns with this paper?

No

Have you any concerns about statistical analyses in this paper?

No

Recommendation?

Major revision is needed (please make suggestions in comments)

Comments to the Author(s)

This paper proposed an effective approach for battery characterization that examines battery performance parameters and the effect of IEEE 802.15.4 based radio load profile is used in this study to estimate the discharge characteristics of multiple battery chemistries. The authors presents a comprehensive investigation and analyse the performance of battery with clear contribution towards the design, discharge characteristic and effect of load currents on battery surface temperatures, however this paper has some minor issues:

1. What is the motivation of proposing this article? These motivations should be highlighted as well as part of the Introduction section.
2. In section 3 (methodology), rewrite/revise the sentence to have a better perspective on battery technology 'no single battery is superior to the others, optimal battery chemistry selection for techno-economic viability.'
3. The authors should explain the significance and features of using 'temperature-controlled chamber' in the experiment. Furthermore, every sentence should be completed or short sentences should be meaningful.
4. For the Section 3-ii, 'Relaxation Test', How the author consider to include the OCV voltages to estimate the optimal duration of the relaxation time?
5. Justify why authors setup the 'discharge pulse at every 15 seconds loaded'?
6. In section 4 (Battery Capacity Test), authors claimed that NiMH exhibited almost 80% flat region for 0.02 and 0.03C discharge rates. The author should add more justification on this result.
7. The authors should add more details about their results in the abstract.
8. The contributions are highlighted well in this paper.
9. Please explain more the details in the results how the battery energy for low-power applications will depend on load and SOC.
10. The language usage throughout this paper needs to be improved, the author should do thorough proofreading on it. Give the article an effective language revision to get rid of few complex sentences that hinder readability and eradicate typo errors.

Decision letter (RSOS-210870.R0)

Dear Dr Ali

The Editors assigned to your paper RSOS-210870 "Battery Characterization for Wireless Sensor Network Applications to Investigate the Effect of Load on Surface Temperatures" have now received comments from reviewers and would like you to revise the paper in accordance with the reviewer comments and any comments from the Editors. Please note this decision does not guarantee eventual acceptance.

Please submit your revised manuscript and required files (see below) no later than 21 days from today's (ie 11-Oct-2021) date. Note: the ScholarOne system will 'lock' if submission of the revision is attempted 21 or more days after the deadline. If you do not think you will be able to meet this deadline please contact the editorial office immediately.

on behalf of Professor Weisi Guo (Associate Editor) and R. Kerry Rowe (Subject Editor)
openscience@royalsociety.org

Reviewer comments to Author:

Reviewer: 1

Comments to the Author(s)

This paper proposed an effective approach for battery characterization that examines battery performance parameters and the effect of IEEE 802.15.4 based radio load profile is used in this study to estimate the discharge characteristics of multiple battery chemistries. The authors presents a comprehensive investigation and analyse the performance of battery with clear contribution towards the design, discharge characteristic and effect of load currents on battery surface temperatures, however this paper has some minor issues:

1. What is the motivation of proposing this article? These motivations should be highlighted as well as part of the Introduction section.
2. In section 3 (methodology), rewrite/revise the sentence to have a better perspective on battery technology 'no single battery is superior to the others, optimal battery chemistry selection for techno-economic viability.'
3. The authors should explain the significance and features of using 'temperature-controlled chamber' in the experiment. Furthermore, every sentence should be completed or short sentences should be meaningful.
4. For the Section 3-ii, 'Relaxation Test', How the author consider to include the OCV voltages to estimate the optimal duration of the relaxation time?
5. Justify why authors setup the 'discharge pulse at every 15 seconds loaded'?
6. In section 4 (Battery Capacity Test), authors claimed that NiMH exhibited almost 80% flat region for 0.02 and 0.03C discharge rates. The author should add more justification on this result.
7. The authors should add more details about their results in the abstract.
8. The contributions are highlighted well in this paper.

9. Please explain more the details in the results how the battery energy for low-power applications will depend on load and SOC.

10. The language usage throughout this paper needs to be improved, the author should do thorough proofreading on it. Give the article an effective language revision to get rid of few complex sentences that hinder readability and eradicate typo errors.

===PREPARING YOUR MANUSCRIPT===

===PREPARING YOUR REVISION IN SCHOLARONE===

<https://royalsociety.org/journals/authors/author-guidelines/#supplementary-material> to include a suitable title and informative caption. An example of appropriate titling and captioning may be found at https://figshare.com/articles/Table_S2_from_Is_there_a_trade-off_between_peak_performance_and_performance_breadth_across_temperatures_for_aerobic_sc_ope_in_teleost_fishes_/3843624.

Author's Response to Decision Letter for (RSOS-210870.R0)

See Appendix A.

RSOS-210870.R1 (Revision)

Review form: Reviewer 1

Is the manuscript scientifically sound in its present form?

Yes

Are the interpretations and conclusions justified by the results?

Yes

Is the language acceptable?

Yes

Do you have any ethical concerns with this paper?

No

Have you any concerns about statistical analyses in this paper?

Yes

Recommendation?

Accept as is

Comments to the Author(s)

In the final version of revision, authors carefully addressed technical comments in the main text. I recommend acceptance on this round.

Decision letter (RSOS-210870.R1)

Dear Dr Ali,

It is a pleasure to accept your manuscript entitled "Battery Characterization for Wireless Sensor Network Applications to Investigate the Effect of Load on Surface Temperatures" in its current form for publication in Royal Society Open Science. The comments of the reviewer(s) who reviewed your manuscript are included at the foot of this letter.

on behalf of Professor Weisi Guo (Associate Editor) and R. Kerry Rowe (Subject Editor)
openscience@royalsociety.org

Reviewer comments to Author:

Reviewer: 1

Comments to the Author(s)

In the final version of revision, authors carefully addressed technical comments in the main text. I recommend acceptance on this round.

Follow Royal Society Publishing on Twitter: [@RSocPublishing](https://twitter.com/RSocPublishing)

Appendix A

Response to Refrees

Updated Files submitted to Editorial Office:

1. Updated manuscript file – “*manuscript_corrections.pdf*”
2. Manuscript track changes – “*track_changes.pdf*”

Response/Actions to reviewer’s comments:

We are very thankful to the esteemed reviewers for their time, efforts, and insightful feedback on our manuscript. The valuable suggestions helped to significantly improve the overall quality of our manuscript. All conceivable efforts have been made to address all the valuable observations and therefore, we would like to present our response measures for the considerations of the review panel.

Reviewer # 1

We are grateful for your time and your extensive input to enhance our manuscript. We have taken the following measures to answer your suggestions.

Comment 1:

What is the motivation of proposing this article? These motivations should be highlighted as well as part of the Introduction section.

Response/action:

A lot of research has been conducted on secondary batteries for grid, micro-grid and electric vehicles (EVs) that require high discharge rates. However, WSN applications require duty-cycled, lower discharge rates that changes the discharge profiles of the batteries. In addition, to the best of our knowledge there are no comprehensive battery discharge datasets available for WSN application. Additionally, to the best of our knowledge, the literature lacked the pre-conditioning methodology for WSN-related batteries. Furthermore, the effect of discharge rate on battery capacity, voltage variations, state-of-charge (SOC) and even the thermal affects are not reported for WSN applications. This served as a motivation to investigate several battery chemistries for WSN-based discharge rates (IEEE 802.15.4 protocol device discharge profiles). The research enabled to observe the behavior of several battery types with variable discharge rates. We also investigated the effects of OCV variations on battery capacity and the finally the thermal effect on battery surfaces. These results are crucial in identifying the most appropriate battery chemistry, as well as the operating conditions for batteries in WSN settings.

As suggested, the motivation is now clearly highlighted in the introduction section and can be tracked at:

(Page 2-3 Lines 19-79)

Comment 2:

In section 3 (methodology), rewrite/revise the sentence to have a better perspective on battery technology ‘no single battery is superior to the others, optimal battery chemistry selection for

techno-economic viability.

Response/action:

We have updated the methodology section by focusing on battery technology, as suggested.

These changes can be tracked at:

(Pages 4-5, Lines 149-155,)

Comment 3:

The authors should explain the significance and features of using ‘temperature-controlled chamber’ in the experiment. Furthermore, every sentence should be completed or short sentences should be meaningful.

Response/action:

The significance of environment testing in controlled environment using a temperature chamber is now discussed in this section. We highlighted the need for temperature-controlled tests, as well as the safety requirements that regulate the need to utilize this equipment in laboratory settings. Furthermore, the short and hanging sentences have been restructured and rewritten to clearly pass the information.

These changes can also be tracked at:

(Pages 5-6, Lines 162-177)

Comment 4:

For the Section 3-ii, ‘Relaxation Test’, How the author consider to include the OCV voltages to estimate the optimal duration of the relaxation time?

Response/action:

The Open Circuit Voltage (OCV) accurately reflect the available battery energy when no load is attached. However, to precisely measure the OCV, a battery should be pre-conditioned to remove the passivation, as well as completely charged. Another important consideration is the relaxation period that is required for the battery to fully re-gain the lost energy due to internal chemical transport phenomenon. A lot of research has been conducted in this perspective, where researchers investigated relaxation times for batteries in order to get accurate OCV measurements. These relaxation periods were reported to be 4-8 hours on average. However, we wanted to address the following changes to gain more clarity on this subject. This included:

1. Relaxation times per battery chemistries
2. Effect on OCV values based on different relaxation times
3. Effect on OCV values due to various SOC discharge levels

This required a new approach to observe the combined effect of various SOC discharge levels, as well as relaxation times on OCV values for several battery chemistries. Our findings revealed that Lithium-based batteries required a relaxation period of (8 hours on average) to fully recover and provide accurate OCV values. Whereas Ni-MH batteries achieved the steady-state within 4 hours. This information is critical in battery conditioning for real-time WSN

applications where inaccurate OCV calculations can result in SOC estimation errors, thus affecting the device lifetime.

We have rewritten the sections to clearly highlight these facts and can be tracked at:

(Page 8, Lines 229-250)

Comment 5:

Justify why authors setup the ‘discharge pulse at every 15 seconds loaded’?

Response/action:

WSN applications commonly require remote sensing, data aggregation, and information transmission. Almost every situation avoids continuous or real-time communication to save energy. As a result, in order to conserve energy and extend device lifetime, WSN nodes are typically operated in duty-cycling mode. As a result, various data transmission rates are generated, each of which is tailored to the intended use. These data transfer rates can be low, medium, high, or near real-time. In these tests, however, we kept the data rate low in order to provide recuperation times between discharge pulses. As a result, a 15-second interval was selected to mimic a low-duty-cycle WSN operational scenario.

These changes can be tracked at:

(Page 6, Lines 186-196)

Comment 6:

In section 4 (Battery Capacity Test), authors claimed that NiMH exhibited almost 80% flat region for 0.02 and 0.03C discharge rates. The author should add more justification on this result.

Response/action:

The flat region as well as lesser voltage variations in Ni-MH batteries are due to a number of internal properties including cell construction, and internal resistance to name a few. The Ni-MH batteries experienced smaller voltage variations and capacity reduction at lower discharge rates, however, at discharge rates of 0.5C and above, a significant reduction is noticed. These changes are highlighted in a sharp “knee” curve in the voltage discharge characteristics.

This phenomenon is explained in detail at:

(Page 9-10, Lines 283-304)

Comment 7:

The authors should add more details about their results in the abstract.

Response/action:

We have completely rewritten the abstract based on the valuable suggestions. The abstract now clearly reflects the need for this research, experiments, and the most important results obtained during the investigations.

Comment 8:

The contributions are highlighted well in this paper

Response/action:

We are very thankful for the insightful comments and suggestions that helped us to improve the quality of our article. We are also very humbled by the kind appreciation by the reviewers.

Comment 9:

Please explain more the details in the results how the battery energy for low-power applications will depend on load and SOC.

Response/action:

The batteries are affected by environmental conditions (such as temperature, humidity) and operational factors (such as discharge rates, and SOC) to name a few. The effect of discharge rates or load directly dictates the amount of time the battery can deliver energy to the load. This corresponds to a shorter battery capacity when the load is increased and vice-versa. On the other hand, SOC reflects the amount of available energy at a certain point in time. For instance, if two identical batteries are discharge using similar loads but with different SOC (such as 100% and 85%), not only does the battery capacity is affected due to the available energy; the discharge profile (the voltage patterns) also varies. For high discharge-rate continuous applications such EVs, the effect could be observed on the battery capacity. Whereas in low power applications, particularly duty cycled WSNs, this translates into device lifetime, and early cut-off levels. We have restructured and rewritten most of the results section and included the suggested points that can be tracked at:

(Page 12-13, Lines 343-378)

Comment 10:

The language usage throughout this paper needs to be improved, the author should do thorough proofreading on it. Give the article an effective language revision to get rid of few complex sentences that hinder readability and eradicate typo errors

Response/action:

We are very grateful for the reviewer's time and efforts in going through all the minute details of our manuscript. This really helped us to restructure the article in a simpler and more effective manner. We have completely re-visited the entire article for grammar, structure, complexity, and typo errors. The updated manuscript was also reviewed by subject matter experts that further helped to add clarity to the article. We believe that our article has improved significantly and therefore we would like to thank the reviewers once again for their valuable suggestions. The supplementary file "*track_changes.pdf*" tracks all these changes, where some of the notable mentions can be tracked at:

(Pages 1-14)